# Home-based cardiac rehabilitation and physical activity in people with heart failure: a secondary analysis of the REACH-HF randomised controlled trials

Grace O Dibben [1,2] Melvyn Hillsdon,[3] Hasnain M Dalal [4,5] Lars H Tang,[6,7] Patrick Joseph Doherty [8] Rod Taylor[1,2]

**Correspondence to**
Dr Grace O Dibben;
Grace.Dibben@glasgow.ac.uk

## ABSTRACT

**Objectives** To quantify the impact of a home-based cardiac rehabilitation intervention (Rehabilitation Enablement in Chronic Heart Failure (REACH-HF)) on objectively assessed physical activity (PA) of patients with heart failure (HF) and explore the extent by which patient characteristics are associated with a change in PA.

**Design** Secondary analysis of randomised controlled trial data.

**Setting** Five centres in the UK.

**Participants** 247 patients with HF (mean age 70.9±10.3 years; 28% women).

**Interventions** REACH-HF versus usual care (control).

**Primary and secondary outcome measures** PA was assessed over 7 days via GENEActiv triaxial accelerometer at baseline (pre-randomisation), post-intervention (4 months) and final follow-up (6–12 months). Using HF-specific intensity thresholds, intervention effects (REACH-HF vs control) on average min/day PA (inactivity, light PA and moderate-to-vigorous PA (MVPA)) over all days, week days and weekend days were examined using linear regression analysis. Multivariable regression was used to explore associations between baseline patient characteristics and change in PA.

**Results** Although there was no difference between REACH-HF and control groups in 7-day PA levels post-intervention or at final follow-up, there was evidence of an increase in weekday MVPA (10.9 min/day, 95% CI: −2.94 to 24.69), light PA (26.9 min/day, 95% CI: −0.05 to 53.8) and decreased inactivity (−38.31 min/day, 95% CI: −72.1 to −4.5) in favour of REACH-HF. Baseline factors associated with an increase in PA from baseline to final follow-up were reduced MVPA, increased incremental shuttle walk test distance, increased Hospital Anxiety and Depression Scale anxiety score and living with a child >18 years (p<0.05).

**Conclusions** While participation in the REACH-HF home-based cardiac rehabilitation intervention did not increase overall weekly activity, patient's behaviour patterns appeared to change with increased weekday PA levels and reduced inactivity. Baseline PA levels were highly predictive of PA change. Future focus should be on robust behavioural changes, improving overall levels of objectively assessed PA of people with HF.

**Trial registration numbers** ISRCTN78539530 and ISRCTN86234930.

## STRENGTHS AND LIMITATIONS OF THIS STUDY

⇒ The use of population-specific intensity thresholds to derive physical activity (PA) intensity provides a more reliable and robust estimate of PA levels of patients with heart failure.

⇒ The use of rigorous data processing and analysis methods to look beyond a single PA metric (ie, average weekly moderate-to-vigorous PA) revealed interesting patterns in the data, however this increase in granularity causes a reduction in statistical power and future studies should consider this in sample size calculations.

⇒ As this was an exploratory study, there may be an increased risk of type 1 errors with multiple repeated tests, and some frequency counts of variables included in multivariable models were low.

## INTRODUCTION

Physical activity (PA) has numerous health benefits for patients with heart failure (HF) including reduced HF mortality and HF hospitalisation, and improved quality of life.[1–3] Current PA guidelines recommend 150 min/week moderate-to-vigorous PA (MVPA) or the equivalent of 30 min/day on 5 or more days/week.[4,5]

Traditionally, cardiac rehabilitation (CR) interventions have prioritised increasing exercise capacity rather than PA behaviour, and evidence that CR increases PA toward recommended levels is lacking.[6] To date, only a small number of studies have assessed the impact of CR on PA in patients with HF, and few results (around 10% of individual results across 10 studies) indicated that CR positively impacts on PA, with studies mainly using self-report measures that are known to be prone to over-reporting.[6,7] Furthermore, in order to derive time spent in MVPA from accelerometers, raw data must be categorised into intensity levels using thresholds or cut-points derived from calibration studies, which have

mostly been undertaken using young, healthy volunteers. Given the increased energy costs of PA in patients with HF, application of these 'generic' intensity thresholds risks misclassifying PA in this population and could make it difficult to tease out small changes in PA behaviour patterns. Because of this, it is unclear whether the lack of positive findings may be due to poor methods of PA measurement in patients with HF, or due to ineffective behaviour change within interventions. New HF-specific accelerometer intensity thresholds for categorising PA intensity from raw accelerometer data have recently been developed, taking into account the lower resting metabolic rate and requirement for greater energy expenditure during PA in people with HF.[8]

Furthermore, although some studies have shown that PA levels are associated with a number of factors including age, body mass index, exercise capacity and disease severity in patients with HF, these have been limited by cross-sectional design that do not measure within-person PA change.[9–11] Exploring the potential patient level characteristics at baseline (ie, socio-demographics, exercise capacity and quality of life) that are associated with later changes in PA level could identify potential subgroups of patients, such as non-responders, who may require more intensive or personalised intervention.

Due to the wide range of health benefits associated with increased PA in patients with HF, further studies, using improved, objective and population-specific PA assessment techniques are needed to understand and clarify the impact of CR on PA levels and the relationships between PA and patient level characteristics. Therefore the primary aim of this study was to assess the impact of a home-based CR programme (Rehabilitation Enablement in Chronic Heart Failure (REACH-HF)[12–15] on objectively measured PA using novel HF-specific accelerometer thresholds for estimating intensity. In addition, we explored the patient level characteristics associated with a change in PA level.

## METHODS

### Patient and public involvement
A key element of the REACH-HF intervention development process was the involvement of a local patient and public involvement (PPI) group consisting of nine patients with HF and caregivers of people with HF. The PPI group co-created the REACH-HF intervention, and were involved in the recruitment process, and associated research questions and topic guides (for more details, see Greaves et al[16]).

### Study design
This secondary analysis used data pooled from two randomised controlled trials (RCTs) that randomised patients with HF 1:1 to a home-based CR intervention plus usual care (REACH-HF group) or to usual care alone (control group), stratified by site and N-terminal Brain Natriuretic Peptide (NT-proBNP). The first was a pilot RCT of the REACH-HF intervention for patients with HF with preserved ejection fraction (HFpEF, left ventricular ejection fraction (LVEF) >45%), and the second was a multicentre RCT of the REACH-HF intervention for patients with HF with reduced ejection fraction (HFrEF, LVEF <45%). Accelerometry-measured PA was a secondary outcome of the original studies, and full details of these trials are presented elsewhere.[12–15]

In summary, REACH-HF is a theory-based, comprehensive self-management programme consisting of four core elements; (1) an HF self-help manual with a choice of two structured exercise programmes; a chair-based exercise DVD, and a progressive walking training programme (patients advised to exercise ≥3 times per week and gradually building in time/distance/walking pace); (2) a patient 'progress tracker' booklet designed to facilitate learning and record symptoms and other actions related to healthcare; (3) a 'family and friends resource' manual for caregivers to increase their understanding of HF and their own physical and mental well-being; and (4) facilitation by specially trained cardiac nurses, physiotherapists or exercise therapists to individually tailor the intervention. Participating patients and caregivers worked through the REACH-HF manual for 12 weeks. Participants in both the pilot RCT and full RCT received the same programme.

This study complies with the Declaration of Helsinki.

### Participants
A total of 247 adults (aged ≥18 years) with a confirmed diagnosis of HF (198 HFrEF, 49 HFpEF), and complete baseline accelerometer data sets (participants were required to have ≥16 hours per day and ≥7 days of wear) were included in this analysis. Participants were recruited and completed the baseline visit between January 2015 and February 2016, from primary and secondary care settings in five UK centres (Birmingham, Cornwall, Dundee, Gwent and York). All participants provided written informed consent.

### Data collection
PA data were collected via accelerometry on three occasions: at baseline (pre-randomisation), post-intervention (4 months) and final follow-up (6–12 months). Given the final accelerometry data follow-up for the two trials was at 6 months for patients with HFpEF, and 12 months for patients with HFrEF we have therefore combined data at these two 'final follow-up' time points for this study.

The following patient level data were collected at baseline: medical history (ie, comorbidities, New York Heart Association (NYHA) class, concomitant medication), socio-demographic information (ie, age, ethnicity, employment status, smoking status), NT-proBNP measurement via blood sample, exercise tolerance via incremental shuttle walk test (ISWT) and health outcome questionnaires, that is, disease-specific health-related quality of life (HRQoL) using the Minnesota Living with Heart Failure Questionnaire, and the Health-Related Quality of Life (HeartQoL) questionnaire; psychological

well-being using the Hospital Anxiety and Depression Scale (HADS) questionnaire; generic HRQoL using the EuroQol-5 Dimension-5 level (EQ-5D-5L) questionnaire; and Self-Care of HF Index questionnaire (SCHFI).

## PA—accelerometry

At the clinical visits, participants were provided with and instructed to wear a GENEActiv triaxial accelerometer (GENEActiv, Activinsights, Kimbolton, Cambridge, UK) for 24-consecutive hours for 7 days. Accelerometers were returned to the clinical trials unit using postage-paid envelopes. Data were downloaded using GENEActiv PC software (V.3.2; Activinsights, Kimbolton, Cambridge, UK) and processed in R (R Core Team, Vienna, Austria) using the GGIR software package (V.1.5–18, http://cran.r-project.org). Initial processing included autocalibration, and detection of abnormally high values and non-wear.[17 18] Data were averaged over 5 s epochs and Euclidean Norm Minus One was used to quantify the acceleration related to movement registered and expressed in milligravity units (mg) using the following formula.[19]

$$\sqrt{x^2 + y^2 + z^2} - 1000\,mg$$

Non-wear was determined over 60 min windows using 15 min increments, and was apparent when two of the three axes had a data range <50 mg and an SD <13 mg.[20] The first consecutive 7 days that met the criteria were used for analysis.

Once the raw accelerometer data was processed, HF-specific accelerometer intensity thresholds (inactivity (ie, <1.5 metabolic equivalents (METs): 16.7 mg (left wrist), 18.6 mg (right wrist), MVPA (ie, ≥3.0 METs): 43.6 mg (left wrist), 45.5 mg (right wrist)) were applied to calculate the average minutes per day spent inactive, in light PA and MVPA, over all days, weekend days and weekdays. These thresholds were established by a recent accelerometer calibration study in 21 patients with HF.[8] Average weekly MVPA was used to calculate the proportion of patients meeting current PA guidelines (≥150 min per week). For the primary analysis these metrics were calculated using bouted data, that is, sustained periods of 10 min or more where accelerometer data lies above the intensity threshold (with a 20% allowance for values to fall outside the threshold). For secondary analyses, these metrics were calculated using unbouted data, that is, allowing PA to be accumulated in bouts of any duration.

## Statistical analysis

Descriptive statistics were calculated as means and SDs or counts and percentages unless otherwise stated.

Data from final follow-up were used for primary analysis in line with the primary endpoints of the two original trials.[12 13] The intervention effects (ie, REACH-HF vs control) on average min/day PA (inactivity, light PA and MVPA) over all days, week days and weekend days were examined using linear regression analysis, adjusting for baseline PA (inactivity, light PA or MVPA, respectively), treatment group and trial stratification variables (NT-proBNP, centre). Secondary analyses included (1) exploring intervention effects at post-intervention follow-up; (2) using unbouted PA data, and (3) examining the proportion of patients meeting PA guidelines were examined using logistic regression.

Univariate linear regression was used to investigate the baseline socio-demographic (eg, age, sex, ethnicity), medical history (eg, NYHA class, medication), ISWT and health status variables (eg, HRQoL) associated with change in MVPA, adjusting for baseline MVPA, treatment group and trial stratification variables (NT-proBNP, centre).

In addition to univariate regression, we undertook multivariable regression to explore the association between selected patient characteristics and change in MVPA. Variables with statistical evidence of univariate association with change in MVPA (p<0.15) were selected for entry into a series of multivariable regression models to establish which variables were independently and most strongly associated with change in PA at final follow-up and post-intervention, mutually adjusting for trial stratification variables, baseline MVPA and treatment group. Model 1—socio-demographic and medical history variables only, model 2—exercise capacity and health status variables only and model 3—socio-demographic, medical history, exercise capacity and health status variables identified as significant (p<0.05) predictors in models 1 and 2. Checks and diagnostics were performed for model assumptions, residuals, multicollinearity (variance inflation factor) and influential observations (Cook's distance). Akaike information criterion (AIC) and $R^2$ values were used to inform model comparison and selection.

Statistical analyses were performed using Stata (V.15.0; StataCorp, College Station, Texas, USA).

## RESULTS

Two hundred and forty-seven patients had accelerometer data at baseline which met the criteria for inclusion in analysis. Post-intervention, 198 patients were included and at final follow-up 173 patients were included in the analysis (figure 1). Patients were predominantly men (72%), had a mean age of 70.9±10.4 years and 60% were NYHA class II (table 1), full baseline characteristics are described in online supplemental table 1. At baseline, the average daily MVPA in REACH-HF group was 43.6 min with 53% participants meeting weekly PA recommendations, and in the control group was 49.7 min per day, with 48% participants meeting weekly PA recommendations.

### Effects of REACH-HF intervention on PA at final follow-up

The effects of REACH-HF intervention versus control at final follow-up on PA (MVPA, light PA and inactivity) over all days, weekdays and weekend days are presented in table 2. Over all 7 days, the average change in daily MVPA at final follow-up in the REACH-HF group was 4.0 min/day, compared with −5.1 min/day in the control group (p>0.05). In both groups, there was huge



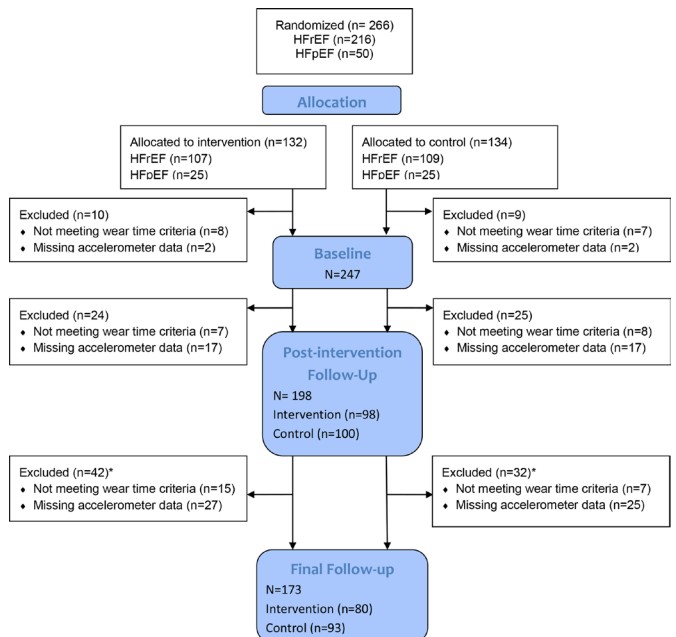

**Figure 1** Participant flow diagram. HFpEF, heart failure with preserved ejection fraction; HFrEF, heart failure with reduced ejection fraction.*14 patients excluded from post-intervention follow-up analysis (for missing data or not enough valid days) were included in the final follow-up with sufficient accelerometer data.

**Table 1** Summary baseline patient characteristics. Data are presented as N (%) unless otherwise stated

| | REACH-HF (N=122) | Control (N=125) |
|---|---|---|
| Mean (SD) age (years) | 70.5 (10.0) | 71.3 (10.7) |
| Female sex | 40 (33) | 30 (24) |
| Mean (SD) BMI (kg/m$^2$) | 29.9 (6.6) | 30.0 (5.9) |
| Ethnicity (white) | 115 (94) | 121 (97) |
| Type of HF (HFpEF) | 24 (20) | 25 (20) |
| NYHA class | | |
| NYHA I | 24 (20) | 16 (13) |
| NYHA II | 71 (58) | 76 (61) |
| NYHA III–IV | 27 (22) | 33 (26) |
| Mean (SD) LVEF (%) | 38.4 (14.7) | 38.1 (15.5) |
| Mean (SD) NT-proBNP (pg/mL) | 1288.3 (1794.3) | 1364.4 (1602.1) |
| Trial site | | |
| Truro | 27 (22) | 29 (23) |
| Gwent | 22 (18) | 22 (18) |
| Birmingham | 24 (20) | 24 (19) |
| York | 25 (20) | 25 (20) |
| Dundee | 24 (20) | 25 (20) |
| Total number of comorbidities* (median, range) | 3 (2–5) | 3 (2–5) |
| Mean (SD) ISWT (peak distance, m) | 241.9 (157.1) | 219.1 (144.2) |
| Mean (SD) MLHFQ | | |
| Overall | 33.5 (24.5) | 30.7 (23.0) |
| Physical | 17.3 (11.8) | 16.1 (11.6) |
| Emotional | 7.7 (7.7) | 7.1 (7.0) |
| Mean (SD) HADS | | |
| Anxiety | 5.3 (4.5) | 5.9 (4.4) |
| Depression | 4.8 (3.6) | 4.8 (3.4) |
| Mean (SD) HeartQoL | | |
| Global | 1.8 (0.8) | 1.8 (0.8) |
| Physical | 1.6 (0.8) | 1.6 (0.8) |
| Emotional | 2.1 (0.9) | 2.1 (0.9) |
| Mean (SD) EQ-5D-5L | 0.7 (0.3) | 0.7 (0.3) |
| Mean (SD) SCHFI | | |
| Maintenance | 56.0 (16.1) | 53.0 (15.4) |
| Management | 42.0 (24.7) | 39.6 (20.4) |
| Confidence | 61.9 (25.2) | 63.6 (23.4) |

*Comorbidities including: angina, diabetes, myocardial infarction, hypertension, osteoporosis, stroke, asthma, chronic back pain, chronic renal impairment, arthritis, atrial fibrillation, chronic obstructive pulmonary disease and depression.
BMI, body mass index; EQ-5D-5L, EuroQol-5 Dimension-5 level; HADS, Hospital Anxiety and Depression Scale; HF, heart failure; HFpEF, heart failure with preserved ejection fraction; HFrEF, heart failure with reduced ejection fraction; ISWT, incremental shuttle walk test; LVEF, left ventricular ejection fraction; MLHFQ, Minnesota Living with Heart Failure Questionnaire; NT-proBNP, N-terminal Brain Natriuretic Peptide; NYHA, New York Heart Association; REACH-HF, Rehabilitation Enablement in Chronic Heart Failure; SCHFI, Self-Care in Heart Failure Index; SD, standard deviation.

variance (minimum −91.2, maximum 291.7 min/day; and minimum −135.7, maximum 173.2 min/day, respectively). Over weekend days, there were no significant between group differences at any intensity, although it appeared that both groups demonstrated similar small reductions in MVPA (−5.0 and −4.6 min/day, respectively) and the REACH-HF group increased their inactivity (11.8 min/day). Over weekdays, there was evidence of an increase in PA over all intensities in the REACH-HF group compared with the control group, however only the increase in light PA and decrease in inactivity reached statistical significance (between group differences in favour of REACH-HF: light PA: 26.87 (95% CI: −0.05 to 53.78), p=0.05, inactivity: −38.31 (95% CI: −72.13 to −4.5), p=0.03).

### Secondary analyses
#### Effects of REACH-HF intervention on PA at post-intervention follow-up
At post-intervention follow-up, there were no significant differences between REACH-HF group and control for all bouted PA intensities or days (all, weekend or week) (online supplemental table 2).

#### Unbouted data
At final follow-up, using unbouted PA data, there was a small increase in weekday MVPA and decrease in weekday inactivity in the REACH-HF group, with the reverse found in control group, (between group differences showing a trend in favour of REACH-HF (MVPA: 15.18 (95% CI: −0.32 to 30.67), p=0.06; inactivity: −21.25

(95% CI: −43.24 to 0.75), p=0.06; online supplemental table 3. The REACH-HF group also appeared to increase their inactivity and decrease PA on weekend days. At post-intervention follow-up there were no significant differences between REACH-HF group and control for

**Table 2** Intervention effects on PA outcomes at final follow-up

| | Baseline | | Final follow-up | | Δ to final follow-up | | Between group difference (mean, 95% CI) p value |
|---|---|---|---|---|---|---|---|
| | REACH-HF (N=80) mean (SD) | Control (N=93) mean (SD) | REACH-HF (N=80) mean (SD) | Control (N=93) mean (SD) | REACH-HF (N=80) mean (SD) | Control (N=93) mean (SD) | |
| **All days bouted** | | | | | | | |
| MVPA (min/day) | 43.62 (51.15) | 49.66 (73.98) | 47.61 (67.78) | 44.56 (72.10) | 3.99 (51.53) | −5.10 (34.23) | 7.61 (−5.11 to 20.33) p=0.24 |
| Light (min/day) | 213.80 (110.94) | 219.93 (114.73) | 221.92 (129.57) | 209.50 (120.90) | 8.12 (87.01) | −10.43 (87.85) | 16.87 (−8.79 to 42.54) p=0.20 |
| Inactive (min/day) | 1182.58 (144.43) | 1170.41 (164.42) | 1170.47 (170.47) | 1185.78 (167.52) | −12.12 (115.83) | 15.53 (10.52) | −24.77 (−56.69 to 7.16) p=0.13 |
| **Weekend days bouted** | | | | | | | |
| MVPA (min/day) | 43.72 (56.62) | 41.0 (69.84) | 38.67 (61.04) | 36.43 (62.10) | −5.05 (52.27) | −4.57 (43.31) | 0.14 (−12.92 to 13.21) p=0.98 |
| Light (min/day) | 208.10 (123.23) | 202.18 (118.25) | 201.36 (128.36) | 206.22 (123.41) | −6.75 (107.42) | 4.04 (103.72) | −8.38 (−37.96 to 21.20) p=0.58 |
| Inactive (min/day) | 1188.18 (161.84) | 1196.82 (160.52) | 1199.98 (162.57) | 1197.35 (158.21) | 11.80 (131.33) | 0.53 (121.78) | 8.27 (−27.01 to 43.55) p=0.64 |
| **Week days bouted** | | | | | | | |
| MVPA (min/day) | 43.58 (50.71) | 53.12 (79.66) | 51.19 (71.70) | 47.82 (78.90) | 7.61 (55.14) | −5.31 (37.46) | 10.87 (−2.94 to 24.69) p=0.12 |
| Light (min/day) | 216.08 (112.68) | 227.03 (120.85) | 230.15 (135.37) | 210.81 (125.37) | 14.07 (90.71) | −16.22 (92.74)* | 26.87 (−0.05 to 53.78) p=0.05 |
| Inactive (min/day) | 1180.34 (143.59) | 1159.85 (174.23) | 1158.66 (178.56) | 1181.37 (178.59) | −21.68 (122.78) | 21.53 (106.40)* | −38.31 (−72.13 to −4.50) p=0.03 |

*P value<0.05 REACH-HF group versus control.

MVPA, moderate-to-vigorous physical activity; PA, physical activity; REACH-HF, Rehabilitation Enablement in Chronic Heart Failure; Δ, change in variable.

all bouted PA intensities or days (all, weekend or week) (online supplemental table 4).

### Prevalence of meeting PA guidelines
In terms of prevalence of meeting PA guidelines, no differences were found between REACH-HF and control group at either follow-up time point (online supplemental tables 5,6).

### Univariate regression analysis
Online supplemental table 7 shows the univariate associations between baseline socio-demographic, clinical and behavioural patient variables and change in MVPA at final follow-up. Older patients were more likely to show a decrease in MVPA (p<0.05), whereas patients living with a child >18 years, with greater ISWT distance and higher HADS anxiety score were more likely to increase their MVPA (p<0.05).

Post-intervention, presence of diabetes and SCHFI maintenance scores at baseline were associated with a decrease in MVPA (p<0.05), whereas individuals with greater ISWT distance at baseline were more likely to increase their MVPA (p<0.05; online supplemental table 8).

### Multivariable regression analysis at final follow-up
Table 3 shows the multivariable prediction models at final follow-up. In model 1: higher baseline MVPA and living with a parent was associated with a decrease in MVPA and living with a child aged >18 years was associated with an increase in MVPA. This model accounted for 15% of the variance in change in MVPA. Three patients were removed with high residual (e=211.5, e=258.3 and e=182.0) and Cook's distance d=0.35, d=0.39 and d=0.04). In model 2: higher baseline MVPA was associated with a decrease in MVPA, and higher ISWT distance and HADS anxiety score were associated with an increase in MVPA. This model accounted for 9% of the variance in MVPA change. In model 3: higher ISWT distance, living with a child aged over 18 years and higher HADS anxiety score were strongly associated with an increase in MVPA and living with a parent and higher baseline MVPA were associated with a decrease in MVPA. The model explained 15% of the variance in MVPA change at final follow-up.

### Secondary analyses
#### Multivariable regression analysis at post-intervention follow-up
Multivariable models to predict change in MVPA post-intervention are presented in online supplemental table 9. In model 1: living with a parent was associated with an MVPA increase, and higher baseline MVPA and presence of diabetes with an MVPA decrease. This model explained 10% of the variance in MVPA change. In model 2: higher baseline MVPA was associated with a decrease in MVPA, and higher ISWT distance was most strongly associated with an increase in MVPA. The model accounted for 10% of the variance in MVPA change. In model 3: living with a parent and higher ISWT distance were associated with MVPA increases, and higher baseline MVPA and presence of



**Table 3** Comparison of multivariable models to predict change in minutes/day MVPA at final follow-up

| Multivariable model | Variables included in model (p<0.05) | Unstandardised beta coefficient (95% CI) | t-statistic | Variable p value | Model adjusted R² (p value) |
|---|---|---|---|---|---|
| 1. Socio-demographic | Group | −4.03 (−12.19 to 4.12) | −0.98 | 0.33 | 0.15 |
| | Baseline MVPA | −0.15 (−0.22 to −0.09) | −4.64 | <0.001 | (<0.001) |
| | Centre | −0.28 (−3.0 to 2.45) | −0.2 | 0.84 | |
| | BNP 2000 | −6.58 (−17.27 to 4.10) | −1.22 | 0.23 | |
| | Live with parent | −41.81 (−72.96 to −10.67) | −2.65 | 0.009 | |
| | Live with child over 18 | 16.04 (0.05 to 32.02) | 1.98 | 0.049 | |
| | constant | 5.89 (−4.92 to 16.70) | 1.08 | 0.28 | |
| 2. Exercise capacity and health status | Group | −6.72 (−19.81 to 6.37) | −1.01 | 0.31 | 0.09 (0.001) |
| | Baseline MVPA | −0.20 (−0.31 to −0.09) | −3.47 | 0.001 | |
| | Centre | 0.13 (−4.34 to 4.60) | 0.06 | 0.95 | |
| | BNP 2000 | −7.39 (−25.08 to 10.30) | −0.83 | 0.41 | |
| | ISWT peak | 0.07 (0.02 to 0.11) | 2.69 | 0.008 | |
| | HADs anxiety | 1.90 (0.41 to 3.38) | 2.52 | 0.013 | |
| | constant | −12.91 (−37.03 to 11.21) | −1.06 | 0.29 | |
| 3. Socio-demographic, exercise capacity and health status* | Group | −8.37 (−21.13 to 4.38) | −1.3 | 0.2 | 0.15 (<0.001) |
| | Baseline MVPA | −0.21 (−0.33 to −0.10) | −3.73 | <0.001 | |
| | Centre | −0.11 (−4.47 to 4.25) | −0.05 | 0.96 | |
| | BNP 2000 | −6.21 (−23.44 to 11.01) | −0.71 | 0.48 | |
| | ISWT peak | 0.08 (0.03 to 0.12) | 3.16 | 0.002 | |
| | Live with child >18 | 30.48 (5.92 to 55.04) | 2.45 | 0.02 | |
| | HADS anxiety | 1.89 (0.44 to 3.33) | 2.58 | 0.01 | |
| | Live with parent | −52.60 (−100.16 to −5.05) | −2.19 | 0.03 | |
| | constant | −14.51 (−38.05 to 9.04) | −1.22 | 0.23 | |

*All variables p<0.05 from multivariate models 1 and 2.
.BNP 2000, NT-proBNP above or below 2000 pg/mL; HADS, Hospital Anxiety and Depression Score; ISWT, incremental shuttle walk test; MVPA, moderate-to-vigorous physical activity.

of diabetes were strongly associated with MVPA decrease. This model accounted for 14% of the variance in MVPA change.

Variance inflation factor (VIF) ranged from 1.06 to 1.18 across the six models indicating a low level of multicollinearity.

## DISCUSSION

We undertook an in-depth analysis of RCT data of a CR intervention (REACH-HF) objectively assessing PA levels of patients with HF using intensity thresholds specifically developed for this population group. When PA levels were averaged over all 7 days of the week, there was no evidence that participation in REACH-HF impacted daily PA levels compared with control. As some 48% of patients in the REACH-HF trials were already meeting UK PA guidelines at baseline (based on bouted data), it may be that there was a ceiling effect, where those already physically active would have difficulty further improving on MVPA levels, making small changes in PA difficult to pick up statistically. This has often been the case in other CR trials.[6]

However, separating weekend and weekdays revealed important patterns in the PA response. Average weekday PA levels showed a consistent trend where MVPA and light PA increased (11 min/day and 27 min/day, respectively), and inactivity decreased (−38 min/day) in the REACH-HF group compared with the control group. Over weekend days, the reverse appeared to be true, with an increase in inactivity and decrease in PA. This pattern was not evident in either group at baseline, therefore these findings suggest that the REACH-HF participants may have compensated for increased PA during the week by being less active at the weekend, which could account for the lack of overall PA increase found. Compensation for increased exercise with increased inactivity in the subsequent days has been demonstrated in other populations such as older adults,[21 22] and overweight/obese people.[23] There are many potential reasons for this, which may or may not be volitional, including perceived fatigue, indirect effects on self-efficacy and motivation and reduction in typical unstructured PA (such as activities of daily living) on non-exercise days.[24] REACH-HF participants were advised to exercise at least three times per week, allowing a day's rest between each session. Participants may have perceived that completion of their exercise sessions during the week meant they could 'reward' themselves with inactivity at the weekend, however qualitative

data would be needed to shed light on the reason for this finding.

Traditionally, CR programmes have been developed with a focus on increasing exercise capacity rather than PA promotion. In its current format we have shown modest positive influences of a home-based CR programme on weekday PA levels of patients with HF. In order to achieve greater PA benefits to patients, CR programmes may need to adapt, placing stronger emphasis on increasing unstructured daily PA behaviour or targeting both PA and sedentary behaviour. This could be facilitated by simplified PA messaging such as sit less, move more, every day, which might be easier for patients to understand and sustain in the longer-term, leading to higher levels of total PA and reduced sedentary time and is likely to have significant health benefits to patients with HF.[25–27] Future interventions may also consider including long-term goal setting or behaviour change coaching.

Given the large variation in PA differences from baseline up to final follow-up in MVPA, we were interested to investigate potential factors associated with a change in MVPA. To our knowledge, this is the first study to explore this. Our results showed that particular living situations of the patient were associated with PA change (living with a parent or child >18 years), which may indicate involvement of family members is associated with PA behaviour change and self-care of patients with HF.[28] Furthermore, the close relationship between exercise capacity and PA level was demonstrated in this study, where baseline MVPA and ISWT distance were the strongest and most consistent factors associated with an increase in PA levels up to 12 months follow-up. This emphasises the importance of individually tailored interventions, where identification of the patient's specific needs or deficits at the start of a CR programme would enable more personalised therapeutic intensity or focus, in order to maximise intervention efficacy.

### Strengths and limitations

This study has a number of strengths. First, the objective measurement of PA, and robust and rigorous accelerometer data processing and analysis. Literature reporting accelerometer measured PA in patients with HF with CR intervention is limited, especially in large, representative samples. The use of population-specific intensity thresholds provides a more reliable estimate of PA levels of patients with HF, compared with application of commonly used thresholds based on healthy adults.[8] However, we acknowledge that these are based on a small heterogeneous sample of patients, and is not a perfect method. Application of a single threshold for MVPA to a population will always lead to a proportion of patients PA being misclassified due to heterogeneity in exercise tolerance. Therefore, further studies are required to find an alternative approach.

This study highlights the benefit of extracting more detailed PA data, looking beyond a single PA metric (ie, average weekly MVPA) and considering within-week differences in PA patterns. In line with updated PA guidelines that acknowledge bouts less than 10 min can also be beneficial for health,[4] we also looked at the unbouted data. We found that removing the bout rule showed that 100% of the participants were meeting PA recommendations. This also provides useful information in patients with HF as performing continuous bouts of exercise for 10 min or more may be challenging with limited exercise tolerance. However, in increasing granularity, statistical power is reduced and future studies should consider this in sample size calculations. Future studies could also consider the distribution of PA both between-days and within-days in patients with HF, since research has shown that afternoon and evening PA decreases with increasing age.[27] This could inform future intervention development, targeting inactive periods throughout both the week and the day best placed for PA modification.

As this was an exploratory study, multiple repeated independent tests were conducted comparing treatment groups and between baseline characteristics and change in MVPA. Given the dangers of multiple testing, which may have led to increased risk of type 1 errors, our results must be treated with caution. In addition, variables associated with MVPA change were inconsistent across the two follow-up points and explained only a small proportion (10–15%) of the variance in MVPA change and combining the final follow-up data sets from 6 and 12 months may have introduced variation to the data. While the sample size was sufficient for this exploratory study, some of the frequency counts of variables included in the multivariable models were low. Further studies using objective PA assessment are needed to clarify the impact of exercise-based CR interventions in patients with HF and the patient factors associated with change in PA.

### CONCLUSION

This pooled analysis of RCTs shows that participation in a home-based CR intervention did not impact 7-day PA compared with control. However, participating in CR did appear to increase weekday PA levels and participants may have compensated for this by becoming more inactive during the weekend. Understanding of the behavioural effects of a home-based CR intervention could provide key targets for clinicians and researchers to adapt CR focus and PA messaging, in order to encourage patients to increase overall weekly PA and reduce inactivity. Understanding the factors associated with change in PA with CR intervention could potentially enhance development of more intense interventions specifically tailored or targeted at those with lower baseline PA or exercise capacity levels, as these patients will have the most to gain from increasing their PA.

**Author affiliations**
[1]MRC/CSO Social and Public Health Sciences Unit, University of Glasgow, Glasgow, UK
[2]College of Medicine and Health, University of Exeter, Exeter, UK
[3]Department of Public Health and Sport Sciences, University of Exeter, Exeter, UK

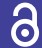

4Research, Development and Innovation, Royal Cornwall Hospitals NHS Trust, TRURO, UK

5Primary Care, University of Exeter Medical School, Truro, UK

6The Research Unit PROgrez, Department of Physiotherapy and Occupational Therapy, Slagelse Hospital, Slagelse, Sjaelland, Denmark

7The Department of Regional Health Research, University of Southern Denmark, Odense, Syddanmark, Denmark

8Department of Health Sciences, University of York, York, UK

**Acknowledgements** The authors thank the patients in the REACH-HF trials and the research team for open access to the trial data. GD completed this research while supported by a University of Exeter Postgraduate Studentship Grant.

**Contributors** GD, RT, MH and HMD were involved in the study conception and interpretation of results. GD and RT led the statistical analysis. GD drafted the manuscript and MH, HMD, LHT, PJD and RT provided comments on the manuscript and signed off the final version. GD is guarantor.

**Funding** This work was supported by a University of Exeter PhD Studentship with support from Professor Taylor's NIHR Senior Investigator Award. Further support was received from the Medical Research Council (grant ref: MC_UU_00022/1) and the Scottish Government Chief Scientist Office (grant ref: SPHSU16).

**Competing interests** RT, HMD, PJD and MH were investigators in the REACH-HF study.

**Patient and public involvement** Patients and/or the public were involved in the design, or conduct, or reporting, or dissemination plans of this research. Refer to the Methods section for further details.

**Patient consent for publication** Not applicable.

**Ethics approval** Ethical approval was granted for the two trials by East of Scotland Research Ethics Service (15/ES/0036) and the North West Lancaster Research Ethics Committee (14/NW/1351). Participants gave informed consent to participate in the study before taking part.

**Provenance and peer review** Not commissioned; externally peer reviewed.

**Data availability statement** Data are available upon reasonable request. This study used secondary randomised controlled trial data that are available through the University of Exeter's Institutional Repository, Open Research Exeter (see https://ore.exeter.ac.uk). Access to these data is permitted but controlled through requests made via the repository to the chief investigator (Professor Taylor: r.s.taylor @ exeter.ac.uk). Although use is permitted, this will be on the basis that the source of the data is acknowledged (including the funder) and it includes reference to the data set 'handle'.

**ORCID iDs**
Grace O Dibben http://orcid.org/0000-0002-7254-5187
Hasnain M Dalal http://orcid.org/0000-0002-7316-7544
Patrick Joseph Doherty http://orcid.org/0000-0002-1887-0237

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
