## [Reviewer comments · BMJ Open]

ARTICLE DETAILS

TITLE (PROVISIONAL)	Home-based cardiac rehabilitation and physical activity in people with heart failure: a secondary analysis of the REACH-HF randomised controlled trials
AUTHORS	Dibben, Grace; Hillsdon, Melvyn; Dalal, Hasnain; Tang, Lars; Doherty, Patrick Joseph; Taylor, Rod

VERSION 1 – REVIEW

REVIEWER	Taniguchi, Tatsunori Osaka University
REVIEW RETURNED	17-May-2022

GENERAL COMMENTS	This paper reports the impact of a home-based cardiac rehabilitation intervention (REACH-HF) on objectively assessed physical activity (PA) of heart failure (HF) patients and explore the extent by which patient characteristics are associated with a change in PA based data pooled from two randomized controlled trials. The authors have addressed an interesting and important topic related to PA in HF. Comments: Abstract: The last sentence in Conclusion section, “Clinicians...”, seems not appropriate description that is derived from the results in this study. Methods: Page 6, Line 18-19. Two trials with different follow-up periods are combined in this study. Please explain the appropriateness of this combination in statistical point of view. Results: Page 9, Line 17-19. Statistical difference between REACH-HF and control group should be provided in the table. Page 9, Line 19-20. Why is the data not shown, while there are so many supplementary tables? Page 10, Line 4-9. Please explain why only the light intensity, bouts PA on weekends was increased instead of change in MVPA?
---

REVIEWER	Giuliano, Catherine Victoria University, Institute for Health and Sport
REVIEW RETURNED	20-Jun-2022

GENERAL COMMENTS

This paper sought to investigate the effects of a home-based cardiac rehabilitation program (REACH-HF) on change in physical activity in patients with heart failure at two time-points; at 4 months and between 6-12 months post intervention. A multivariable regression also explored associations between baseline characteristics and change in physical activity. This paper explores an important and relevant research question for the field of cardiac rehabilitation. Participation in REACH-HF did not increase weekly PA, but tended to increase weekday (Monday to Friday) activity at 6-12 months follow-up and decreased physical inactivity.

Introduction: The introduction refers to the small number of studies that have assessed the impact of CR on PA levels. It would be valuable to add the results of these studies and then discuss the limitations, which will help strengthen the rationale for this study. It would be valuable to expand on the novelty of the method used and the value added with this approach (ie. HF specific PA thresholds). The rationale for the multivariate analysis could be further strengthened. The study objectives are clearly stated. Please change all references to "HF patients" to "patients with HF" throughout the article

Methods: The patient and public involvement statements requires further development. Authors can find detailed guidance on what to include in this section in the author guidelines. The study design chosen is appropriate for the research question, however, the methods section is not described in a systematic way that would enable easy replication and is not easy to follow. At times the descriptions confuse (or merge) the methods from the original studies with the current study. For example, in several places authors refer to primary and secondary analyses or outcomes, sometimes referring to the original REACH-HF study (eg. Page 8 line 33) and other times referring to the current study (eg. Page 5 line 25). Another example on page 5 line 41-46 "The primary outcome was change in disease-specific HR-WOL with accelerometer assessed PA collected as a secondary outcome". Authors should refer to primary and secondary analyses with reference to this current study only. The participants sections also refers to participants from the original study but without mention of inclusion or exclusion for this current study, which differs from the original.

Results: Whilst the outcomes are defined well, the results section does not read easily due to the number of outcomes reported. For instance, the results section reports bouted and unbouted data at 2 time points, and also across weekdays, weekends only and total week days. Whilst it may be appropriate to include all (which the authors have done in the text as well as in supplementary data), it is difficult for the reader to easily follow the results in the format currently written. Authors should consider reducing outcomes reported in detail in the text and providing in supplementary data only, or using subheadings to help organise the structure. The discussion should also follow this logic.

The multivariate regression analysis section requires more detail for some of the variables. For example, it states "baseline MVPA...was associated with an increase in MVPA". Does this mean a higher or lower MVPA was associated with an increase? This needs to be stated for all baseline variables which showed an association eg, HADS score, ISWT.

	Discussion: The discussion explores the findings of this study well. The suggestion that patients may compensate for increased PA during the week by being less active on the weekend requires further exploration. Why would patients do this? Do you think this is conscious? Were patients already more active during the week on baseline PA patterns? Is this actually a change to PA patterns across the week? Some further concepts to explore: what other improvements could be made to CR programs to achieve behaviour change? Did REACH-HF involve long term goal setting or behaviour change coaching? Do future programs need to consider adding these modules? If fitter and more active patients are more likely to improve, what can we do to support the others?
--	---

REVIEWER	Vasankari, Sini University of Turku
REVIEW RETURNED	19-Nov-2022

GENERAL COMMENTS	The subject in question is interesting, and applying objective measurement of physical activity in this patient group is important for more precise cardiac rehabilitation. While this is an exploratory study, the sample size was quite large. In addition, including patient group specific intensity thresholds increases the reliability for physical activity classification. A few suggestions to make the tables clearer for the reader: Table 1: The presentation of ethnicity and type of HF are a bit unclear. What does the number refer to? Table 3: Could the N be next to the heading (e.g. REACH-HF (N=80)), and the proportion of patients provided with percentage in parenthesis (e.g. 42 (53%))? Also, a tiny change: p.8 line 10: "Average weekly MVPA (≥ 150 minutes per week) was used to calculate the proportion of patients meeting current PA guidelines." The parenthesis should be at the end of the sentence as it describes the guideline. In this study, both unbouted and bouted (threshold of 10 min for physical activity) analyses have been used. Unlike older PA recommendations, the new ones also acknowledge bouts less than 10 minutes beneficial for health. It can be seen in the results that including the threshold has a crucial effect on the proportion meeting the guidelines. I find this interesting, and it could also be brought up in the discussion.
---

VERSION 1 – AUTHOR RESPONSE

Reviewer 1:

Comment	Response
This paper reports the impact of a home-based cardiac rehabilitation intervention (REACH-HF) on objectively assessed physical activity (PA) of heart failure (HF) patients and explore the extent by which	Thank you.

patient characteristics are associated with a change in PA based data pooled from two randomized controlled trials. The authors have addressed an interesting and important topic related to PA in HF.	
Abstract: The last sentence in Conclusion section, "Clinicians...", seems not appropriate description that is derived from the results in this study.	We have edited this sentence to be more appropriate.
Methods: Page 6, Line 18-19. Two trials with different follow-up periods are combined in this study. Please explain the appropriateness of this combination in statistical point of view.	We accept the reviewers comment, and have clarified this in the methods section. This has also been raised as a potential limitation of the study.
Results: Page 9, Line 17-19. Statistical difference between REACH-HF and control group should be provided in the table.	We have not performed statistical tests for baseline differences between REACH-HF and control groups, as this is discouraged by CONSORT. http://www.bmj.com/content/340/bmj.c869
Page 9, Line 19-20. Why is the data not shown, while there are so many supplementary tables?	Instances of "data not shown" have been removed from the manuscript.
Page 10, Line 4-9. Please explain why only the light intensity, bouted PA on weekends was increased instead of change in MVPA?	Text edited to state that there was evidence of higher MVPA in REACH-HF vs control, but this did not reach significance.

Reviewer #2

Comment	Response
Reviewer: 2 This paper sought to investigate the effects of a home-based cardiac rehabilitation program (REACH-HF) on change in physical activity in patients with heart failure at two time-points; at 4 months and between 6-12 months post intervention. A multivariable regression also explored associations between baseline characteristics and change in physical activity. This paper explores an important and relevant research question for the field of cardiac rehabilitation. Participation in REACH-HF did not increase weekly PA, but tended to increase weekday (Monday to Friday) activity at 6-12 months follow-up and decreased physical inactivity.	Thank you
Introduction: The introduction refers to the small number of studies that have assessed the impact of CR on PA levels. It would be valuable to add the results of these studies and then discuss the limitations, which will	Thank you for these suggestions, we have made the following changes:  The synthesised vote-counting results of the small number of HF studies

help strengthen the rationale for this study. It would be valuable to expand on the novelty of the method used and the value added with this approach (ie. HF specific PA thresholds). The rationale for the multivariate analysis could be further strengthened. The study objectives are clearly stated. Please change all references to “HF patients” to “patients with HF” throughout the article	included in the previous review have been added to the discussion  • The limitations of previous PA measurement methods and the value of our approach have been discussed in more detail in the introduction. • We have added further text to the introduction and methods section regarding the multivariable analysis. • All instances of HF patients have been changed to patients with HF as advised.
Methods: The patient and public involvement statements requires further development. Authors can find detailed guidance on what to include in this section in the author guidelines. The study design chosen is appropriate for the research question, however, the methods section is not described in a systematic way that would enable easy replication and is not easy to follow. At times the descriptions confuse (or merge) the methods from the original studies with the current study. For example, in several places authors refer to primary and secondary analyses or outcomes, sometimes referring to the original REACH-HF study (eg. Page 8 line 33) and other times referring to the current study (eg. Page 5 line 25). Another example on page 5 line 41-46 “The primary outcome was change in disease-specific HR-WOL with accelerometer assessed PA collected as a secondary outcome”. Authors should refer to primary and secondary analyses with reference to this current study only. The participants sections also refers to participants from the original study but without mention of inclusion or exclusion for this current study, which differs from the original.	We agree with the reviewer and have made the following changes:  • We have added more detail to the PPI statement, and provided a reference to the intervention development publication. • We have made edits to the text throughout the methods section to improve the readability and avoid confusion between the original trials, and the current study.
Results: Whilst the outcomes are defined well, the results section does not read easily due to the number of outcomes reported. For instance, the results section reports bouts and unbouted data at 2 time points, and also across weekdays, weekends only and total week days. Whilst it may be appropriate to include all (which the authors have done in the text as well as in supplementary data), it is difficult for the reader to easily follow the results in the format currently written. Authors should consider reducing outcomes reported in detail in the	We agree with the reviewer and have made the following changes:  • The results section text has been edited, we have removed one of the tables from the main text and added it to the supplementary files and added subheadings to try and improve the readability and clarity for the reader. • The multivariate regression analysis section has been edited to clarify the direction of associations as recommended.

text and providing in supplementary data only, or using subheadings to help organise the structure. The discussion should also follow this logic. The multivariate regression analysis section requires more detail for some of the variables. For example, it states “baseline MVPA...was associated with an increase in MVPA”. Does this mean a higher or lower MVPA was associated with an increase? This needs to be stated for all baseline variables which showed an association eg, HADS score, ISWT.	
Discussion: The discussion explores the findings of this study well. The suggestion that patients may compensate for increased PA during the week by being less active on the weekend requires further exploration. Why would patients do this? Do you think this is conscious? Were patients already more active during the week on baseline PA patterns? Is this actually a change to PA patterns across the week? Some further concepts to explore: what other improvements could be made to CR programs to achieve behaviour change? Did REACH-HF involve long term goal setting or behaviour change coaching? Do future programs need to consider adding these modules? If fitter and more active patients are more likely to improve, what can we do to support the others?	Thank you, we have made the following changes to the discussion:  • We have explored potential reasons for our PA pattern findings. • We have also explored some of the further concepts/future CR program advice as recommended.

Reviewer #3

Comment	Response
The subject in question is interesting, and applying objective measurement of physical activity in this patient group is important for more precise cardiac rehabilitation. While this is an exploratory study, the sample size was quite large. In addition, including patient group specific intensity thresholds increases the reliability for physical activity classification.	Thank you
Table 1: The presentation of ethnicity and type of HF are a bit unclear. What does the number refer to?	The tables have been updated to clarify ethnicity and type of HF.
Table 3: Could the N be next to the heading (e.g. REACH-HF (N=80)), and the proportion of patients provided with percentage in parenthesis (e.g. 42 (53%))?	We have moved the N and put the percentages in brackets as recommended for Table 3 and all tables included in the supplementary files.

Also, a tiny change: p.8 line 10: "Average weekly MVPA (≥ 150 minutes per week) was used to calculate the proportion of patients meeting current PA guidelines." The parenthesis should be at the end of the sentence as it describes the guideline.	The parenthesis has been moved as recommended.
In this study, both unbouted and bouted (threshold of 10 min for physical activity) analyses have been used. Unlike older PA recommendations, the new ones also acknowledge bouts less than 10 minutes beneficial for health. It can be seen in the results that including the threshold has a crucial effect on the proportion meeting the guidelines. I find this interesting, and it could also be brought up in the discussion.	Thank you, we have included a comment about this in the discussion.

VERSION 2 – REVIEW

REVIEWER	Vasankari, Sini University of Turku
REVIEW RETURNED	01-Jan-2023
GENERAL COMMENTS	The authors have responded to all comments and revised the manuscript accordingly. I find the manuscript ready for publication.